# REGION-ADAPTIVE SAMPLING FOR DIFFUSION TRANSFORMERS

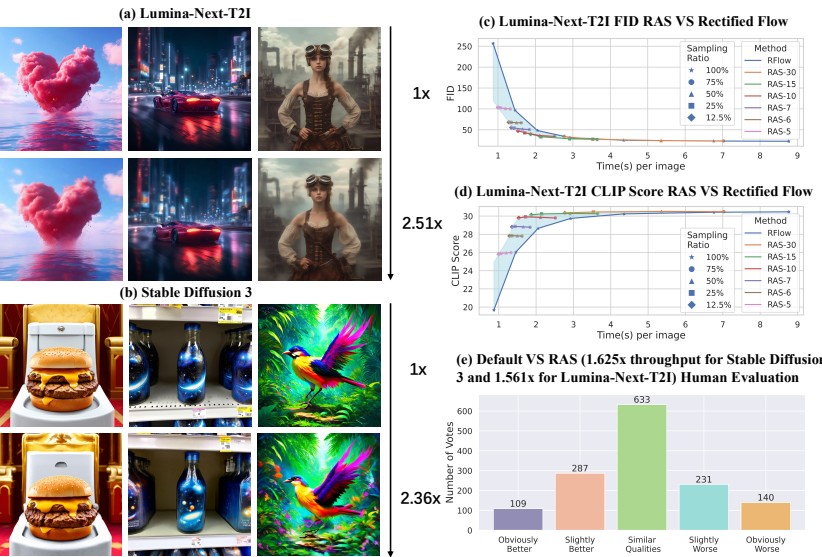

Figure 1: (a)(b) Accelerating Lumina-Next-T2I and Stable Diffusion 3, with 30 and 28 steps separately. (c)(d) Multiple configurations of RAS outperform rectified flow in both image qualities and text-following. RAS-X stands for RAS with X sampling steps in total. (e) RAS achieves comparable human-evaluation results with the default model configuration while achieving around 1.6x speedup.

## ABSTRACT

Diffusion models (DMs) have become the state-of-the-art for generative tasks across domains, but their reliance on sequential forward passes limits real-time performance. Prior acceleration methods mainly reduce sampling steps or reuse intermediate results. Leveraging the flexibility of Diffusion Transformers (DiTs) to handle variable token counts, we propose *RAS*, a training-free sampling strategy that dynamically assigns different update ratios to image regions based on model focus. Our key observation is that at each step, DiTs concentrate on semantically meaningful areas, and these regions exhibit strong continuity across consecutive steps. Exploiting this, *RAS* updates only focused regions while reusing cached noise for others, with focus determined from the previous step's output. Evaluated on Stable Diffusion 3 and Lumina-Next-T2I, *RAS* achieves up to 2.36× and 2.51× speedups, respectively, with minimal quality loss. This demonstrates a practical step toward more efficient diffusion transformers for real-time generation.

## 1 INTRODUCTION

Diffusion models (DMs) Ho et al. (2020); Dhariwal & Nichol (2024); Song & Ermon (2019); Sohl-Dickstein et al. (2015) have proven to be highly effective probabilistic generative models, producing high-quality data across various domains. Applications of DMs include image synthesis Rombach et al. (2022); Dhariwal & Nichol (2021), image super-resolution Li et al. (2022); Yue et al.

(2024); Gao et al. (2023), image-to-image translation Wang et al. (2022); Saharia et al. (2022); Li et al. (2023), image editing Kawar et al. (2023); Zhang et al. (2023), inpainting Lugmayr et al. (2022), video synthesis Blattmann et al. (2023); Esser et al. (2023), text-to-3D generation Poole et al. (2022), and even planning tasks Janner et al. (2022). However, generating samples with DMs involves solving a generative Stochastic or Ordinary Differential Equation (SDE/ODE) Protter & Protter (2005); Hartman (2002) in reverse time, which requires multiple sequential forward passes through a large neural network. This sequential processing limits their real-time applicability.

Considerable work has been dedicated to accelerating the sampling process in DMs by reducing the number of sampling steps. Approaches include training-based methods such as progressive distillation Salimans & Ho, consistency models Song et al. (2023), and rectified flow Liu et al. (2022); Albergo & Vanden-Eijnden (2022); Lipman et al. (2022), and training-free methods such as DPM-solver Lu et al. (2022), AYS Sabour et al. (2023), Deep-Cache Xu et al. (2018), and Delta-DiT Chen et al. (2024b). These methods uniformly process all regions of an image during sampling, irrespective of the specific needs of different regions. Intuitively, however, the complexity of

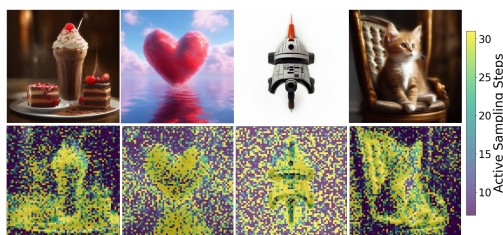

Figure 2: The main subject and the regions with more details are *brushed* for more steps than other regions in *RAS*. Each block represents a patchified latent token.

different regions within an image varies: intricate foreground elements may require more sampling steps for clarity, while repetitive backgrounds could benefit from more aggressive compression of sampling steps without significant loss of quality. This suggests a potential for a more flexible sampling approach that can dynamically adjust the sampling ratio across different regions, enabling faster, yet high-quality diffusion process.

This concept is a natural progression in the evolution of DMs. From DDPM Ho et al. (2020) to Stable Diffusion XL Podell et al. (2023), diffusion models have predominantly relied on U-Nets, whose convolutional structures Ronneberger et al. (2015) necessitate uniform treatment of all image regions due to fixed square inputs. However, with the advent of DiTs Peebles & Xie (2023) and the increasing exploration of fully transformer-based architectures Vaswani (2017), the research focus has shifted towards architectures that can accommodate flexible token inputs , opening up new possibilities. This shift has inspired us to design a new sampling approach capable of assigning different sampling steps to different regions within an image.

To assess the feasibility of this idea, we visualized diffusion outputs at different sampling steps (Figure 3). Two patterns emerged: (1) regions of focus show strong continuity across adjacent steps in later stages, and (2) each step concentrates on semantically meaningful areas of the image. This resembles an artist refining a canvas in thousands of strokes, where each step selectively improves certain regions. Consequently, areas ignored at a given step could be skipped in DiT computation, allowing resources to focus on regions of interest.

We validated this hypothesis by ranking tokens at each step using our proposed output-noise metric, which highlights regions of primary focus. Measuring ranking similarity with NDCG (Figure 4) revealed high continuity between adjacent steps, motivating a sampling strategy that allocates different ratios to regions based on their attention persistence.

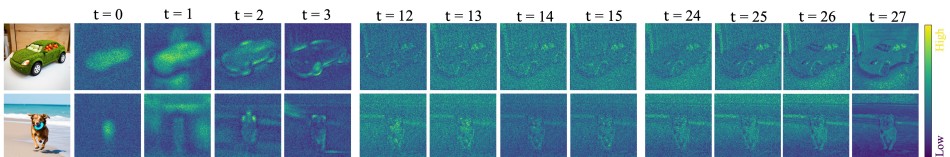

Figure 3: Visualization of predicted noise of each step. DiT model focuses on certain regions during each step and the change in focus is continuous across steps.

As is shown in Figure 5, our method leverages the output noise from the previous step to identify the model's primary focus for the current step (fast-update regions), allowing only these regions to

proceed through DiT for denoising. Conversely, for regions of less interest (slow-update regions), we reuse the previous step's noise output directly. This approach enables regional variability in sampling steps: areas of interest are updated with a higher ratio, while others retain the previous noise output, thus reducing computation.

For each input $X_t$, we select a fast-update rate to determine the regions needing updates in each step, while regions in the slow-update regions retain the previous noise output, which, combined with the updated fast-region noise, forms $X_{t-1}$ for the next step. To maintain global consistency, we keep features from slow-update regions as reference keys and values for subsequent steps. Although the fast-region selection is dynamic and recalculated after each update to prioritize significant areas, we periodically reset the inference for all regions to mitigate cumulative errors.

In summary, we propose *RAS*, the first diffusion sampling strategy that allows for regional variability in sampling ratios. Compared to spatially uniform samplers, this flexibility enables our approach to allocate DiT's processing power to the model's current areas of interest, signifi-

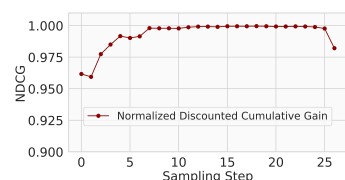

Figure 4: NDCG Järvelin & Kekäläinen (2000); Wang et al. (2013) for each pair of adjacent sampling steps is high throughout the diffusion process, marking the similarities in the ranking of focused tokens ranging from 0 to 1.

cantly improving generation quality within the same inference budget. As shown in Figure 1 (c)(d), our method achieves substantial reductions in inference cost with minimal FID increase, while outperforming the uniform sample baseline in terms of FVD within equivalent inference times. Figure 1 (a)(b) also demonstrates that with models like Lumina-Next-T2I andå Le Zhuo et al. (2024) and Stable Diffusion 3 Esser et al. (2024b), our method's fast-region noise updating yields over twice the acceleration with minimal image quality loss.

## 2 RELATED WORK

### 2.1 DIFFUSION MODELS: FROM U-NET TO TRANSFORMER

Diffusion models Ho et al. (2020); Dhariwal & Nichol (2024); Song & Ermon (2019); Sohl-Dickstein et al. (2015) have shown strong generative capabilities, often surpassing GANs Goodfellow et al. (2014) in downstream tasks. Early approaches such as DDPMs Ho et al. (2020) and Stable Diffusion XL Podell et al. (2023) primarily relied on convolutional U-Nets Ronneberger et al. (2015). However, convolutional backbones require preserving spatial resolution for operations like pooling, limiting the ability to exploit redundancy in latent inputs and making pruning difficult.

This limitation has been addressed by Diffusion Transformers (DiTs) Peebles & Xie (2023), now adopted in state-of-the-art models including Stable Diffusion 3 Esser et al. (2024a), Lumina T2X andå Le Zhuo et al. (2024), and Pixart-Sigma Chen et al. (2024a). Unlike U-Nets, DiTs use a pure Transformer architecture Vaswani (2017) with adaptive layer norm for conditional prompts, eliminating convolution entirely. Positional information is provided via embeddings, making latent tokens independent of spatial constraints. This independence allows us to exploit redundancy (Section 1) by computing only the most relevant tokens at each step while caching others' noise predictions from previous steps.

### 2.2 EFFICIENT DIFFUSION MODEL INFERENCE

To address the problem of high inference cost in diffusion models, various acceleration techniques have been proposed from different perspectives. A commonly used approach is to reduce the number of sampling steps. Some of these techniques require additional training, such as progressive distillation Salimans & Ho, consistency models Song et al. (2023), and rectified flow Liu et al. (2022); Lipman et al. (2022); Albergo & Vanden-Eijnden (2022). Among these methods, rectified flow has been widely used in models like Stable Diffusion 3 Esser et al. (2024a). It learns the ODE to follow straight paths between the standard normal distribution and the distribution of the training dataset. These straight paths significantly reduce the distance between the two distributions, which in turn lowers the number of sampling steps needed.

Training-free methods have also been proposed to reduce either the number of sampling steps or the per-step computation. For example, DeepCache Xu et al. (2018), tailored for U-Net-based models, caches and retrieves features across adjacent stages to skip certain down- and up-sampling operations. However, such methods treat all image regions uniformly, overlooking the varying complexity across different parts of an image and leading to inefficiency.

As discussed in Section 1, image regions often differ substantially in complexity. To exploit this heterogeneity, we propose *RAS*, which optimizes computation by adapting processing to region-specific characteristics. *RAS* is orthogonal to prior techniques—such as step reduction or module-level optimizations (e.g., DiTFastAttn Yuan et al. (2024) and $\Delta$-DiT Chen et al. (2024b))—and can be combined with them for further efficiency.

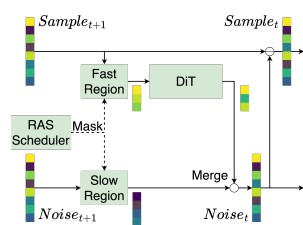

Figure 5: Overview of *RAS* design. Only current fast-update regions of each step are passed to the model.

## 3 METHODOLOGY

### 3.1 OVERVIEW

In this section, we present the *RAS* design and techniques to exploit inter-timestep token correlations and the regional token attention mechanism introduced in Section 1. (1) Based on the regional characteristics we observed in the DiT inference process, we propose an end-to-end pipeline that dynamically eliminates the computation through DiT of certain tokens at each timestep. (2) To leverage the continuity across consecutive timesteps, we propose a

Table 1: Meanings of the symbols that are used in this paper

| | |
|---|---|
| $t$ | The current timestep |
| $N$ | The noise output of the DiT model |
| $\widetilde{N}$ | The cached noise output from the previous timestep |
| $\hat{N}$ | The estimated full-length noise calculated with $N$ and $\widetilde{N}$ |
| $S$ | The unpathified image sample |
| $x$ | The pathified input of the DiT model |
| $M$ | Mask generated to drop certain tokens in the input |
| $D$ | The number of times the tokens in a patch being dropped |

straightforward method to identify the fast-update regions that require refinement in upcoming timesteps. (3) Building on our observations of continuous distribution patterns, we introduce several scheduling optimization techniques to further enhance the quality of generated content.

### 3.2 REGION-ADAPTIVE SAMPLING

**Region-Aware DiT Inference with *RAS*.** Building on the insight that only certain regions are important at each timestep, we introduce the *RAS* pipeline for DiT inference. In U-Net-based models such as SDXL Podell et al. (2023), tokens must remain in fixed positions to preserve positional information. However, given the structure of DiT, we can now mask and reorder elements within latent samples, as positional information is already embedded using techniques like RoPE Su et al. (2024). This flexibility allows us to selectively determine which regions are processed by the model. To achieve this, some additional operations are required starting from the final step. At the end of each timestep, the current sample is updated by combining the fresh model output for the active tokens and the cached noise for the inactive tokens. Specifically, the noise for the entire sequence is restored by integrating both the model output and the cached noise from the previous step. This mechanism enables active, important tokens to move in the new direction determined at the current timestep, while the inactive tokens retain the trajectory from the previous timestep. We then compute the metric $R$, which is used to identify the fast-update regions based on the noise, update the drop count $D$ to track the frequency with which each token has been excluded, and generate the mask $M$ accordingly. With the mask $M$, the noise for the slow-update regions is cached, while the sample for the current fast-update regions is patchified and passed through the DiT model. Since modules like Layernorm and MLP do not involve cross-token operations, the computation remains unaffected even when the sequence is incomplete. For the attention Vaswani

(2017) module, we introduce a caching mechanism to further enhance performance, which will be detailed later. In summary, *RAS* dynamically detects regions of focus and reduces the overall computational load of DiT by at least the same proportion as the user-defined sampling ratio.

**Region Identification.** The DiT model processes the current timestep embedding, latent sample, and prompt embedding to predict the noise that guides the current sample closer to the original image at each timestep. To quantify the refinement of tokens at each timestep, we use the model's output as a metric. Through observation, we found that the standard deviation of the noise strongly marks the regions in the images, with the main subject (fast-update regions) showing an obvious lower standard deviation than the background (slow-update region). This could be caused by the difference in the amount of information between the regions after mixing with the Gaussian noises. Utilizing the deviation as a metric achieves reasonable results of image qualities and notable differences between regions, as is shown in Figure 7. Also, considering the similarities between latent samples across adjacent timesteps, we hypothesize that tokens deemed important in the current timestep are likely to remain important in the next, while the less-focused tokens can be dropped with minimal impact. Before we reach the final formulation of the metric, we need to introduce another technique to prevent starvation.

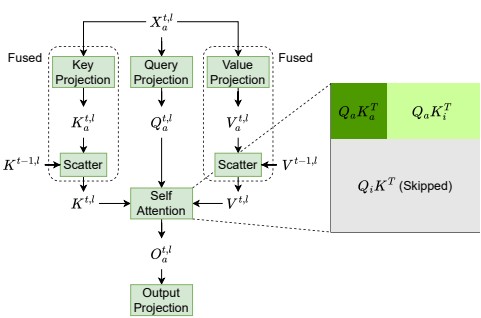

Figure 6: A *RAS* self-attention module using Attention Recovery to enhance generation quality. $X_a^{t,l}$, $Q_a^{t,l}$, $K_a^{t,l}$, $V_a^{t,l}$ and $O_a^{t,l}$ represent the input hidden states, query, key, value and attention output of active tokens on layer $l$ during step $t$, respectively. $K^{t,l}$ and $V^{t,l}$ denote the key and value caches. The scatter operation to partially upload the key and value caches are fused into the previous projection using a PIT GeMM kernel. The keys and values of the not-focused area ($K_i^{t,l}$ and $V_i^{t,l}$) are estimated with the cache from the last sampling step ($K^{t-1,l}$ and $V^{t-1,l}$).

**Starvation Prevention.** During the diffusion process, the main subject regions typically require more refinement compared to the background. However, consistently dropping computations for background tokens can lead to excessive blurring or noise in the final generated image. To address this, we track how often a token is dropped and incorporate this count as a scaling factor in our metric for selecting tokens to cache or drop, ensuring less important tokens are still adequately processed.

Additionally, since DiT patchifies the latent tokens before feeding them into the model, we compute our metric at the patch level by averaging the scores of the tokens within each patch. Combining all the factors mentioned above, our metric can be written as:

$$R_t = mean_{patch}(std(\hat{N}_t)) \cdot exp(k * D_{patch}) \tag{1}$$

where $\hat{N}_t$ is the current estimated noise, $D_{patch}$ is the count of how many times the tokens in a patch have been dropped, and k is a scale factor to control the difference of sample ratios between fast-update regions and slow-update regions.

**Key and Value Caching.** As we know, the attention mechanism works by using the query for each token to compute its attention score with each other tokens by querying the keys and values of the whole sequence, thus giving the relations between each two tokens. The attention of the active tokens in *RAS* can be calculated with only other active tokens. However, the metric **R** we introduce to identify the current fast and slow regions does not take their contribution to the attention score into consideration. Thereby, losing these tokens during attention can cause a huge change in the final output. Our solution here is also caching. During each step, the full keys and values are cached until they are partially updated with the current active tokens. As is described in Figure 6, this solution is also based on the similarity between each two sampling steps, and now we can estimate the original attention output by:

$$O_a = softmax(\frac{Q_a[K_a, \widetilde{K}_i]^T}{\sqrt{d}})[V_a, \widetilde{V}_i] \tag{2}$$

where $i$ stands for the inactive tokens.

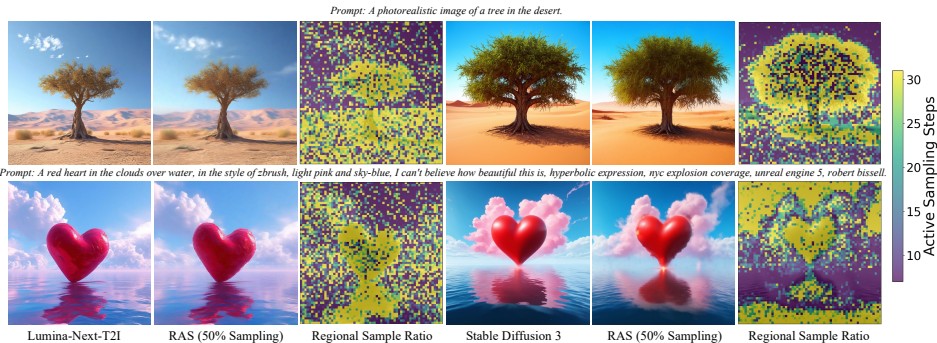

Figure 7: Visualization of *RAS* on Lumina-Next-T2I and Stable Diffusion 3.

## 3.3 SCHEDULING OPTIMIZATION

**Dynamic Sampling Ratio.** As shown in Figure 4, correlations between timesteps are lower in the early stages but increase as the diffusion process stabilizes, consistent with the patterns in Figure 3. This indicates that applying selective sampling too early could harm the structural foundation of the generated image. To account for this, we adopt a dynamic sampling strategy: the first few steps (e.g., 4 out of 28) use a full 100% ratio to preserve image outlines, after which the ratio is gradually reduced during the stable phase. This design balances efficiency and quality, enabling substantial computational savings while minimizing negative effects on the final output.

**Accumulated Error Resetting.** *RAS* focuses on the model's regions of interest, which tend to be similar across adjacent sampling steps. However, regions that are not prioritized for multiple steps may accumulate stale denoising directions, resulting in significant error between the original latent sample and the one generated with *RAS*. To mitigate this issue, we introduce dense steps into the *RAS* diffusion process to periodically reset accumulated errors. For instance, in a 30-step diffusion process where *RAS* is applied starting from step 4, we designate steps 12 and 20 as dense steps. During these dense steps, the entire image is processed by the model, allowing it to correct any drift that may have developed in unfocused areas. This approach ensures that the accumulated errors are reset, maintaining the denoising process in alignment with the correct direction.

Table 2: Pareto Improvements of rectified flow with *RAS* on COCO Val2014 1024×1024. Full experiment results are available in Figure 2 and the Supplementary Material.

| Method | Steps | Sample Ratio | Image/s↑ | FID ↓ | sFID ↓ | CLIP↑ score |
|---|---|---|---|---|---|---|
| **SD3** | | | | | | |
| RFlow | 5 | 100% | 1.43 | 39.70 | 22.34 | 29.84 |
| RAS | 7 | 25.0% | 1.45 | **31.99** | 21.70 | **30.64** |
| RAS | 7 | 12.5% | 1.48 | 32.86 | 22.10 | 30.55 |
| RAS | 6 | 25.0% | 1.52 | 33.24 | **21.51** | 30.38 |
| RAS | 6 | 12.5% | **1.57** | 33.81 | 21.62 | 30.33 |
| RFlow | 4 | 100% | 1.79 | 61.92 | 27.42 | 28.45 |
| RAS | 5 | 25.0% | 1.94 | **51.92** | **25.67** | **29.06** |
| RAS | 5 | 12.5% | **1.99** | 53.24 | 26.04 | 28.94 |
| **Lumina** | | | | | | |
| RFlow | 7 | 100% | 0.49 | 48.19 | 38.60 | 28.65 |
| RAS | 10 | 25.0% | 0.59 | **45.67** | **32.36** | **29.82** |
| RAS | 10 | 12.5% | **0.65** | 47.34 | 32.69 | 29.75 |
| RFlow | 5 | 100.% | 0.69 | 96.53 | 59.26 | 26.03 |
| RAS | 7 | 25.0% | 0.70 | **53.93** | **39.80** | **28.85** |
| RAS | 7 | 12.5% | 0.74 | 54.62 | 40.23 | 28.83 |
| RAS | 6 | 25.0% | 0.75 | 67.16 | 46.46 | 27.85 |
| RAS | 6 | 12.5% | **0.78** | 67.88 | 45.88 | 27.83 |

## 3.4 IMPLEMENTATION

**Kernel Fusing.** As previously mentioned, we introduced key and value caching in the self-attention mechanism. In each attention block of the selective sampling steps, these caches are partially updated by active tokens and then used as key and value inputs for the attention functions. This partial updating operation is equivalent to a scatter operation with active token indices.

In our scenario, the source data of the scatter operation comprises active keys and values outputted by the previous general matrix multiplication (GeMM) kernel in the linear projection module. The extra GPU memory read/store on active keys and values can be avoided by fusing the scatter operation into the GeMM kernel, rather than launching a separate scatter kernel. Fortunately, PIT Zheng et al. (2023) demonstrates that all permutation invariant transformations, including one-dimensional scattering, can be performed in the I/O stage of GPU-efficient computation kernels (e.g. GeMM kernels) with minimal overhead. Using this method, we fused the scatter operation into the epilogue of the previous GeMM kernel.

Table 3: Memory Consumption of RAS.

| Stable Diffusion 3 | | | |
|---|---|---|---|
| Method | Steps | Memory (GB) | Speedup |
| RFlow | 28 | 19.21 (1x) | 1x |
| RAS-50% | 28 | 20.36 (1.06x) | 1.62x |
| RAS-12.5% | 28 | 20.36 (1.06x) | 2.44x |

| Lumina-Next-T2I | | | |
|---|---|---|---|
| Method | Steps | Memory (GB) | Speedup |
| RFlow | 30 | 10.30 (1x) | 1x |
| RAS-50% | 30 | 10.73 (1.04x) | 1.56x |
| RAS-12.5% | 30 | 10.73 (1.04x) | 2.70x |

## 4 EXPERIMENTS

### 4.1 EXPERIMENT SETUP

**Models, Datasets, Metrics and Baselines.** We evaluate *RAS* on Stable Diffusion 3 Esser et al. (2024a) and Lumina-Next-T2I andå Le Zhuo et al. (2024) for text-to-image generation tasks, using 10,000 randomly selected caption-image pairs from the MS-COCO 2017 dataset Lin et al. (2014). To assess the quality of generated images and their compatibility with prompts, we use the Fréchet Inception Distance (FID) Heusel et al. (2017), the Sliding Fréchet Inception Distance (sFID) Heusel et al. (2017), and the CLIP score Hessel et al. (2021) as evaluation metrics. For baseline comparison, we evaluate *RAS* against widely-used Rectified-Flow-based Flow-Matching methods Liu et al. (2022); Albergo & Vanden-Eijnden (2022); Esser et al. (2024a); Lipman et al. (2022); Dao et al. (2023); Fischer et al. (2023), which uniformly reduce the number of timesteps in the generation process for the whole image.

**Code Implementation.** We implement *RAS* using PyTorch Paszke et al. (2019), leveraging the diffusers library von Platen et al. (2022) and its FlowMatchEulerDiscreteScheduler. The evaluation metrics are computed using public repositories available on GitHub Seitzer (2020); Hu (2022); Zhengwentai (2023). Experiments are conducted on four servers, each equipped with eight NVIDIA A100 40GB GPUs, while speed tests are performed on an NVIDIA A100 80GB GPU.

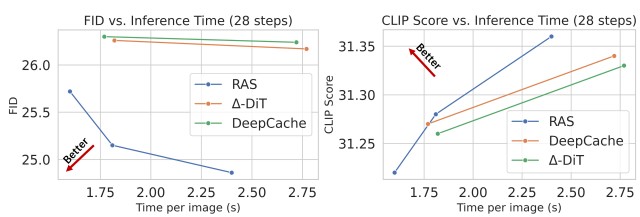

Figure 8: Comparison on Stable Diffusion 3 of RAS with DeepCache and Δ-DiT, utilizing different cache interval.

### 4.2 GENERATION BENCHMARKS

We conducted a comparative evaluation of *RAS* and the rectified flow, which uniformly reduces the number of timesteps for every token during inference. To assess the performance of *RAS*, we performed experiments using various configurations of inference timesteps. The findings can be interpreted in two principal ways.

**Pushing the Efficiency Frontier.** From the first aspect, *RAS* offers a chance to further reduce the inference cost for each number of timesteps rectified flow offers. As illustrated in Figure 1 (c)(d), we generated 10,000 images using dense inference across different timesteps, ranging from 3 to 30. Subsequently, we applied *RAS* at varying average sampling ratios over selective sampling timesteps, with a set of timesteps. The results indicate that *RAS* can significantly reduce inference time while exerting only a minor effect on key evaluation metrics.

Furthermore, the efficiency improvements achieved with *RAS* are attained at a lower cost compared to merely reducing the number of timesteps. Specifically, the rate of quality degradation observed when decreasing the sampling ratio of *RAS* is considerably lower than that observed when reducing the number of timesteps in dense inference, particularly when the number of timesteps is fewer than 10. This demonstrates that *RAS* constitutes a promising approach to enhancing efficiency while maintaining output quality and ensuring compatibility with prompts.

Table 4: Comparison with detailed prompts on ParaImage-3000Wu et al. (2023a). R-X% represents RAS with X% sample ratio.

| Method | FID ↓ | sFID ↓ | CLIP ↑ | time/image (s) |
|---|---|---|---|---|
| RFlow | 36.54 | 40.25 | 34.29 | 3.90 |
| R-75% | 37.24 | 40.23 | 34.18 | 3.05 |
| R-50% | 38.96 | 41.17 | 34.12 | 2.40 |
| R-25% | 40.82 | 41.41 | 34.00 | 1.81 |
| R-12.5% | 42.13 | 40.25 | 33.96 | 1.59 |

**Pareto Improvements of Uniform Sampling.** We observed that *RAS* often yields Pareto improvements for rectified flow. To illustrate this, we sorted results from Stable Diffusion 3 and Lumina-Next-T2I by throughput and compared different *RAS* configurations with the closest baselines in Table 2. Across nearly all cases, *RAS* achieves higher throughput while simultaneously improving FID, sFID, and CLIP scores over dense rectified-flow inference. This demonstrates that for any given throughput level, *RAS* not only offers configurations with both superior speed and quality, but also expands the parameter space for balancing efficiency, fidelity, and prompt alignment.

### 4.3 MEMORY CONSUMPTION

As *RAS* requires caching the intermediate noise and the corresponding keys and values during inference, we evaluate the extra memory consumption of *RAS* in Table 3. RAS requires 6% and 4% extra memory respectively with Stable Diffusion 3 and Lumina-Next-T2I, which is acceptable compared with the speedup. Also, the extra memory does not vary with the sample ratio as the whole activations are cached for later usage.

### 4.4 COMPARISON WITH LAYER-WISE METHODS

Although orthogonal, we compare *RAS* with widely-used layer-wise

Table 5: Benchmarks for evaluating the human preference on Lumina-Next-T2X. RAS-X% stands for RAS with X% tokens activated each step. RAS provides Pareto improvements in multiple settings.

| Method | Steps | Time(s) | SpeedUp ↑ | Img. Rew. ↑ | PickScore ↑ | hpsv2 ↑ |
|---|---|---|---|---|---|---|
| RFlow | 30 | 8.77 | 1 | 0.37 | 21.88 | 0.26 |
| RFlow | 15 | 4.36 | 2.01 | 0.13 | 21.45 | 0.24 |
| RAS-25% | 30 | 3.89 | 2.26 | **0.13** | **21.45** | 0.22 |
| RAS-75% | 15 | 3.72 | 2.35 | 0.05 | 21.34 | **0.24** |
| RFlow | 10 | 2.92 | 3 | -0.20 | 20.94 | 0.21 |
| RAS-25% | 15 | 2.31 | 3.78 | **-0.18** | **20.98** | **0.21** |
| RFlow | 7 | 2.05 | 4.27 | -0.75 | 20.24 | 0.19 |
| RAS-25% | 10 | 1.70 | 5.15 | **-0.43** | **20.54** | **0.19** |
| RAS-12.5% | 10 | 1.54 | 5.68 | **-0.54** | **20.34** | 0.18 |

cached-based methods for better comprehension on a subset of 5000 images from COCO. We manually adapted DeepCacheXu et al. (2018) for DiT by reusing features and reproduced Δ-DiTChen et al. (2024b) according to its paper. As Figure 8 shows, *RAS* achieves greater speedup while improving FID and CLIP scores.

### 4.5 DETAILED PROMPTS, OBJECTS, POSITIONS AND COUNTS.

To evaluate the effect of *RAS* in scenarios when using extremely detailed prompts, and when the user requires exact numbers or positions of the objects, we test *RAS* on the ParaImage-3000 Wu et al. (2023a) and GenEvalGhosh et al. (2023) dataset. Results show that RAS has little effect on the overall score and provides Pareto improvement in multiple fields. Please find the detailed results in the Appendix.

### 4.6 HUMAN EVALUATION

To assess whether *RAS* improves throughput while preserving quality, we conducted a human evaluation. We sampled 14 prompts from the official papers and blogs of Stable Diffusion 3 and Lumina, generating two images per prompt: one with dense inference and one with *RAS*, using the same

random seed and timesteps. During the selective sampling period, *RAS* used a 50% average sampling ratio. We recruited 100 participants from 18 universities and companies to compare the paired outputs.

As shown in Figure 1(e), 45.21% of 1400 votes judged the two images to be of similar quality, while 28.29% favored the dense result and 26.50% preferred *RAS*. These results indicate that *RAS* achieves substantial throughput gains (1.625× on Stable Diffusion 3 and 1.561× on Lumina-Next-T2I) with negligible impact on human preference.

Furthermore, we evaluate *RAS* on ImageReward Xu et al. (2023), PickScore Kirstain et al. (2023), and hpsv2 Wu et al. (2023b), which are commonly used for assessing human preferences. As is shown in Figure 5, *RAS* achieves high performance on the benchmarks while providing higher speed.

### 4.7 ABLATION STUDY

**Token Drop Scheduling.** As shown in Table 6 (a), we evaluate the

Table 6: Ablation Study on Stable Diffusion 3. All techniques including dynamic sampling ratio, region identifying, error reset, key & value recovery are necessary for high quality generation.

(a) Drop Scheduling

| Method | FID ↓ | sFID ↓ | CLIP score ↑ |
|---|---|---|---|
| Default | 35.81 | 18.41 | 30.13 |
| Static Sampling Freq. | 37.92 | 19.11 | 29.98 |
| Random Dropping | 43.19 | 22.23 | 29.65 |
| W/O Error Reset | 46.10 | 24.85 | 30.41 |

(b) Key and Value Caching

| Method | Timesteps | FID ↓ | sFID ↓ | CLIP score ↑ |
|---|---|---|---|---|
| Default | 28 | 24.30 | 26.26 | 31.34 |
| W/O | 28 | 31.36 | 20.19 | 31.29 |
| Default | 10 | 35.81 | 18.41 | 30.13 |
| W/O | 10 | 32.33 | 20.21 | 30.27 |

(c) Error Reset Schedule

| Reset ID | FID ↓ | sFID ↓ | CLIP ↑ |
|---|---|---|---|
| 5 | 27.04 | 19.03 | 31.33 |
| 8 | 24.60 | 17.24 | 31.31 |
| 11 | 25.80 | 16.67 | 31.17 |
| 7,11 | 24.58 | 15.82 | 31.31 |

(d) Starvation Prevention

| Method | Steps | FID ↓ | sFID ↓ | CLIP ↑ |
|---|---|---|---|---|
| Default | 10 | 35.81 | 18.41 | 30.13 |
| W/O | 10 | 39.87 | 19.75 | 29.84 |
| Default | 14 | 26.48 | 18.14 | 31.18 |
| W/O | 14 | 26.58 | 17.96 | 31.11 |

scheduling configurations introduced in Section 3, including sampling ratio scheduling, selection of cached tokens, and the insertion of dense steps during the selective sampling period to reset accumulated errors, using 10 timesteps with an average sampling ratio of 12.5% on Stable Diffusion 3. The results indicate that each of these techniques contributes to the overall quality of *RAS*.

**Key and Value Caching.** As shown in Table 6 (b), caching keys and values from the previous step is crucial, especially when generating high-quality images with more timesteps. While dropping the keys and values of non-activated tokens during attention can improve throughput, it significantly affects the attention scores of activated tokens. A token's low ranking in the model output does not necessarily mean it has no contribution to the attention scores of other tokens.

**Error resetting schedule.** As is shown in Table 6(c), we conducted experiments on the schedule of error resetting with 14 steps on Stable Diffusion 3. Results show that inserting an error resetting set in the middle of the RAS process (from step 4 to 13) provides the best performance. Inserting more dense steps provides little improvement compared with the extra time overhead.

**Starvation Prevention**. Table 6(d) proves the necessity of starvation prevention, which brings no obvious extra overhead.

## 5 CONCLUSION

In this paper, we proposed *RAS*, a novel diffusion sampling strategy that dynamically adjusts sampling rates according to regional attention, thereby allocating computational resources more efficiently to areas of greater importance Extensive experiments and user studies demonstrate that *RAS* achieves substantial speed-ups with minimal degradation in quality, outperforming uniform sampling baselines and paving the way for more efficient and adaptive diffusion models.

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

# A APPENDIX

## A.1 THE USE OF LARGE LANGUAGE MODELS

This paper only uses Large Language Models to polish the writing and grammar.

## A.2 EVALUATION OF DETAILED PROMPTS, OBJECTS, POSITIONS, AND COUNTS.

| Method | Step | Sing. Obj. ↑ | Two Obj. ↑ | Count ↑ | Color ↑ | Pos. ↑ | SpeedUp ↑ | Overall Score ↑ |
|---|---|---|---|---|---|---|---|---|
| RFlow | 30 | 0.92 | 0.44 | 0.40 | 0.70 | 0.08 | 1 | 0.45 |
| RAS-25% | 30 | **0.92** | **0.44** | 0.39 | 0.69 | 0.07 | 1.25 | 0.44 |
| RAS-50% | 30 | **0.92** | 0.41 | **0.40** | 0.68 | 0.08 | 1.56 | 0.44 |
| RFlow | 15 | 0.91 | 0.39 | 0.37 | 0.67 | 0.07 | 2.01 | 0.42 |
| RAS-75% | 30 | **0.91** | 0.37 | **0.37** | **0.67** | **0.07** | 2.25 | **0.42** |
| RAS-87.5% | 30 | 0.89 | 0.33 | 0.35 | **0.67** | 0.05 | 2.70 | 0.40 |

Table 7: GenEval of RAS and RFlow on Lumina. GenEval evaluates the method's ability to follow instructions, including single object, two objects, object counting, colors, and positions, and gives an overall score. RAS poses little effect on the overall score while provides high speedup.

| Method | FID ↓ | sFID ↓ | CLIP ↑ | time/image (s) |
|---|---|---|---|---|
| RFlow | 36.54 | 40.25 | 34.29 | 3.90 |
| R-75% | 37.24 | 40.23 | 34.18 | 3.05 |
| R-50% | 38.96 | 41.17 | 34.12 | 2.40 |
| R-25% | 40.82 | 41.41 | 34.00 | 1.81 |
| R-12.5% | 42.13 | 40.25 | 33.96 | 1.59 |

Table 8: Comparison with detailed prompts on ParaImage-3000Wu et al. (2023a). R-X% represents RAS with X% sample ratio.

To evaluate the effect of *RAS* in scenarios when using extremely detailed prompts, and when the user requires exact numbers or positions of the objects, we test *RAS* on the ParaImage-3000 Wu et al. (2023a) and GenEvalGhosh et al. (2023) dataset, which evaluates the model's ability to generate single, two, multiple objects, colors, and positions with a fixed set of prompts and gives an overall score. As is shown in Table 7 and 8, RAS has little effect on the overall score and provides Pareto improvement in multiple fields.

## A.3 MORE VISUALIZATION OF *RAS*

This section presents *RAS* accelerating Lumina-Next-T2I and Stable Diffusion 3 with a 50% sampling ratio. As illustrated in Figure 10, the main object receives more sampling steps compared to the background, demonstrating the significance of our region-adaptive sampling strategy. This approach ensures that the primary subject in the generated image consistently undergoes more sampling, while relatively smooth regions receive fewer sampling steps. For instance, in the example shown in Figure 10 with the prompt "hare in snow," the weeds in the snow are sampled more frequently, while the smooth snow receives fewer sampling steps.

In Figure 11, we visualize the standard deviation of the noise across dimensions, as well as the decoded images derived from the noise. This stems from our observation that the noise's standard deviation is consistently smaller in the main subject areas. A preliminary hypothesis is that this occurs because the main subject contains more information. When mixed with a certain proportion of noise at each diffusion step, the foreground tends to retain more deterministic information compared to the background. This allows the model to predict more consistent denoising directions. We acknowledge that further study is needed to fully understand this phenomenon.

The primary contribution of this work is to highlight that employing different sampling steps for different regions can significantly enhance the efficiency of diffusion model sampling. The method for selecting these regions is not limited to the aforementioned approach based on the noise standard

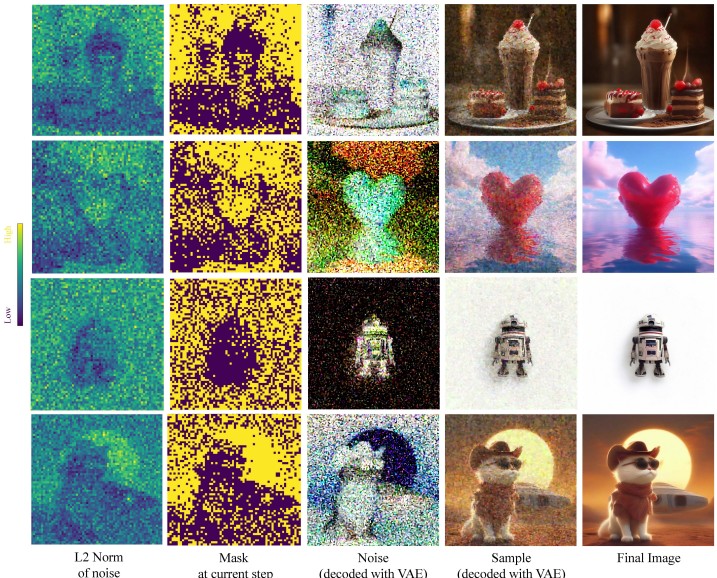

L2 Norm of noise  Mask at current step  Noise (decoded with VAE)  Sample (decoded with VAE)  Final Image

Figure 9: *RAS* using norm as the metric, accelerating Lumina-Next-T2I with 50% sample ratio and 30 total steps. The noise, masks and samples are from the 20th step.

deviation across dimensions. For example, we also experimented with using the $l-2$ norm of the noise output by the network as a criterion for selection. By targeting regions with larger noise norms, which indicate areas the network deems requiring more refinement, we observed a preference for more complex regions in the frequency domain as in Figure 9. This approach also achieves high-quality imaging results, as shown in Table 9. It can be seen that the methods using the $l_2$ norm and standard deviation (std) yield relatively similar results, and both significantly outperform random selection, particularly when the cache ratio is higher.

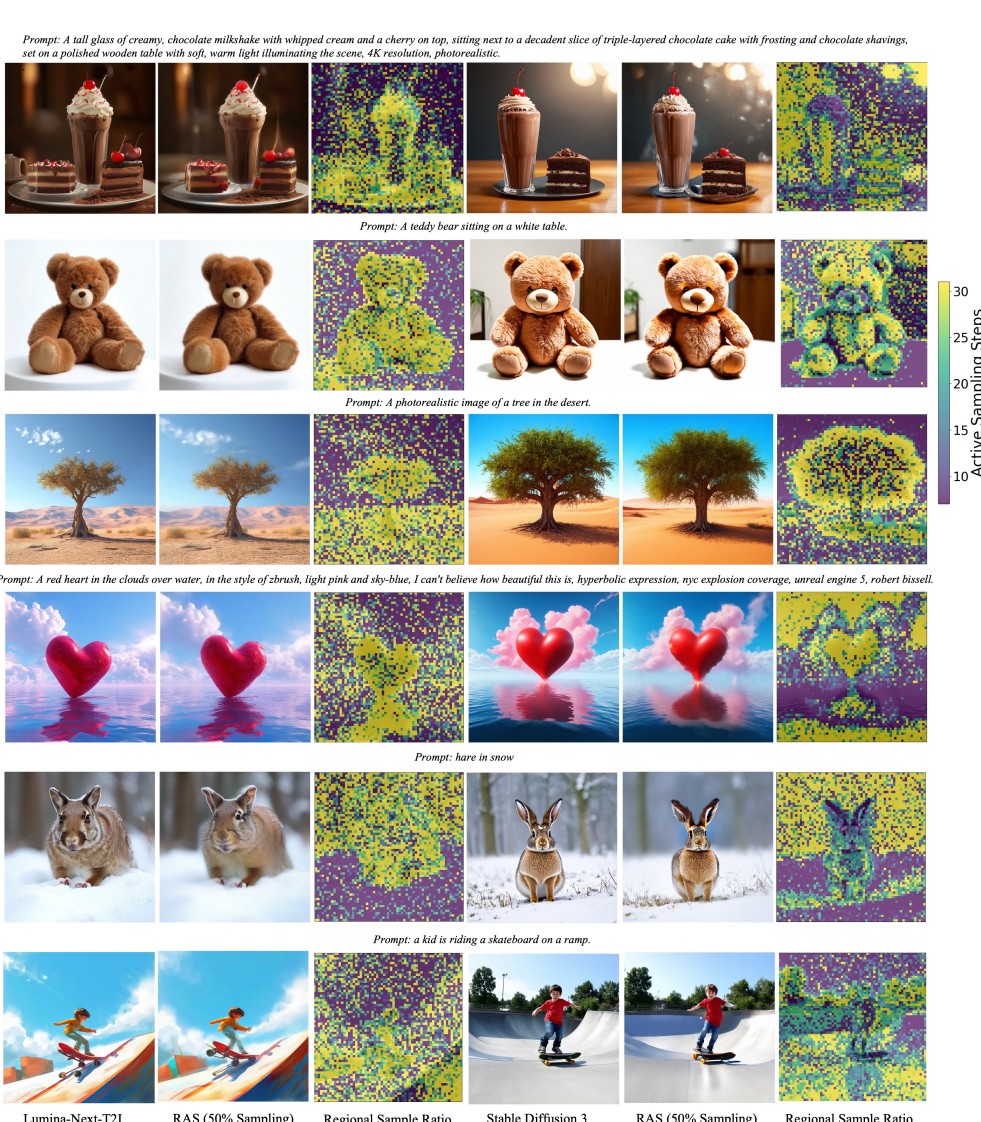

Figure 10: *RAS* VS default sampling and the active sampling step for each latent token.

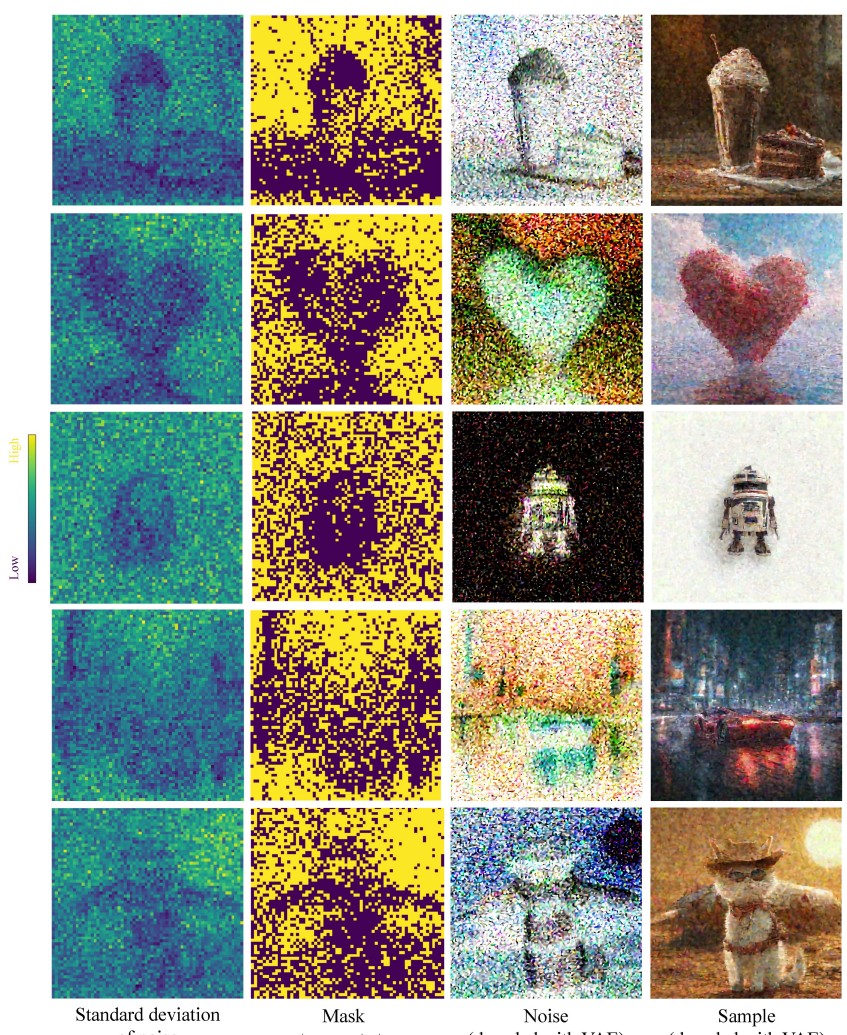

Figure 11: The 20th sampling step (out of 30) of Lumina-Next-T2I using *RAS*.

| Method | Sample Steps | Sampling Ratio | Image/s ↑ | FID ↓ | sFID ↓ | CLIP score ↑ |
|---|---|---|---|---|---|---|
| RFlow | 7 | 100.0% | 1.01 | 27.23 | 17.76 | 30.87 |
| RAS-Std | 7 | 25.0% | 1.45 | 31.99 | 21.7 | 30.64 |
| RAS-Norm | 7 | 25.0% | 1.45 | 31.65 | 21.24 | 30.59 |
| Random | 7 | 25.0% | 1.45 | 33.26 | 22.10 | 30.67 |

Table 9: Experiments on using L2 Norm as the metric for *RAS* on Stable Diffusion 3. The sample ratio of the first 4 steps is 100% to guarantee generation qualities.

## A.4 FULL EXPERIMENT RESULTS OF *RAS*

In this section, we present the full experiment results of *RAS* against rectified flow, with the same settings as is described in the experiment section. Both Table 10 and 11 are ordered by the throughputs.

| Method | Sample Steps | Sampling Ratio | Image/s ↑ | FID ↓ | sFID ↓ | CLIP score ↑ |
|---|---|---|---|---|---|---|
| RFlow | 30 | 100.0% | 0.11 | 22.46 | 16.59 | 30.47 |
| RAS | 30 | 75.0% | 0.14 | 23.31 | 17.73 | 30.49 |
| RFlow | 23 | 100.0% | 0.15 | 23.10 | 17.91 | 30.42 |
| RAS | 30 | 50.0% | 0.18 | 24.10 | 18.83 | 30.51 |
| RFlow | 15 | 100.0% | 0.23 | 24.88 | 21.02 | 30.25 |
| RAS | 30 | 25.0% | 0.26 | 27.44 | 20.95 | 30.45 |
| RAS | 15 | 75.0% | 0.27 | 26.82 | 23.33 | 30.26 |
| RAS | 30 | 12.5% | 0.31 | 33.64 | 23.44 | 30.36 |
| RAS | 15 | 50.0% | 0.33 | 28.48 | 25.17 | 30.29 |
| RFlow | 10 | 100.0% | 0.34 | 31.35 | 27.84 | 29.74 |
| RAS | 10 | 75.0% | 0.40 | 34.19 | 30.57 | 29.79 |
| RAS | 15 | 25.0% | 0.43 | 33.28 | 27.41 | 30.24 |
| RAS | 15 | 12.5% | 0.48 | 39.75 | 28.88 | 30.14 |
| RAS | 10 | 50.0% | 0.48 | 36.18 | 32.36 | 29.86 |
| RFlow | 7 | 100.0% | 0.49 | 48.19 | 38.60 | 28.65 |
| RAS | 7 | 75.0% | 0.54 | 50.45 | 40.19 | 28.78 |
| RAS | 10 | 25.0% | 0.59 | 42.96 | 33.51 | 29.91 |
| RAS | 7 | 50.0% | 0.61 | 51.78 | 40.51 | 28.82 |
| RAS | 6 | 75.0% | 0.62 | 66.12 | 46.58 | 27.80 |
| RAS | 10 | 12.5% | 0.65 | 47.34 | 32.70 | 29.75 |
| RAS | 6 | 50.0% | 0.67 | 66.54 | 46.71 | 27.83 |
| RAS | 7 | 25.0% | 0.70 | 53.93 | 39.80 | 28.85 |
| RAS | 7 | 12.5% | 0.74 | 54.62 | 40.23 | 28.83 |
| RAS | 6 | 25.0% | 0.74 | 67.16 | 46.46 | 27.85 |
| RAS | 5 | 75.0% | 0.75 | 99.01 | 56.26 | 26.02 |
| RAS | 6 | 12.5% | 0.78 | 67.88 | 45.89 | 27.83 |
| RFlow | 5 | 100.0% | 0.69 | 96.53 | 59.26 | 26.03 |
| RAS | 5 | 50.0% | 0.83 | 99.81 | 56.57 | 26.01 |
| RAS | 5 | 25.0% | 0.95 | 101.50 | 56.40 | 25.93 |
| RAS | 5 | 12.5% | 1.00 | 102.90 | 55.25 | 25.84 |
| RFlow | 3 | 100.0% | 1.15 | 256.90 | 94.80 | 19.67 |

Table 10: Full experiment results of *RAS* and rectified flow on Lumina-Next-T2I and COCO Val2014 1024×1024.

| Method | Sample Steps | Sampling Ratio | Image/s ↑ | FID ↓ | sFID ↓ | CLIP score ↑ |
|---|---|---|---|---|---|---|
| RFlow | 28 | 100% | 0.26 | 25.8 | 15.32 | 31.4 |
| RAS | 28 | 75.0% | 0.33 | 24.43 | 15.94 | 31.39 |
| RAS | 28 | 50.0% | 0.42 | 24.86 | 16.88 | 31.36 |
| RFlow | 14 | 100% | 0.51 | 24.49 | 14.78 | 31.34 |
| RAS | 28 | 25.0% | 0.55 | 25.16 | 17.11 | 31.29 |
| RFlow | 12 | 100% | 0.59 | 24.36 | 14.89 | 31.3 |
| RAS | 14 | 75.0% | 0.62 | 23.61 | 15.92 | 31.35 |
| RAS | 28 | 12.5% | 0.63 | 25.72 | 17.3 | 31.22 |
| RFlow | 10 | 100% | 0.71 | 24.17 | 15.39 | 31.22 |
| RAS | 14 | 50.0% | 0.74 | 24.6 | 17.24 | 31.32 |
| RAS | 14 | 25.0% | 0.91 | 25.88 | 17.97 | 31.24 |
| RAS | 10 | 75.0% | 0.91 | 24.39 | 16.29 | 31.12 |
| RAS | 14 | 12.5% | 0.98 | 26.48 | 18.14 | 31.18 |
| RAS | 10 | 50.0% | 1.0 | 27.1 | 17.5 | 30.93 |
| RFlow | 7 | 100% | 1.01 | 27.23 | 17.76 | 30.87 |
| RAS | 7 | 75.0% | 1.16 | 27.57 | 18.76 | 30.81 |
| RAS | 10 | 25.0% | 1.2 | 30.97 | 18.36 | 30.67 |
| RAS | 10 | 12.5% | 1.3 | 35.81 | 18.41 | 30.13 |
| RAS | 7 | 50.0% | 1.3 | 30.04 | 20.34 | 30.73 |
| RAS | 6 | 75.0% | 1.3 | 31.23 | 19.98 | 30.48 |
| RAS | 6 | 50.0% | 1.41 | 32.21 | 20.86 | 30.43 |
| RFlow | 5 | 100% | 1.43 | 39.7 | 22.34 | 29.84 |
| RAS | 7 | 25.0% | 1.45 | 31.99 | 21.7 | 30.64 |
| RAS | 7 | 12.5% | 1.48 | 32.86 | 22.1 | 30.55 |
| RAS | 6 | 25.0% | 1.52 | 33.24 | 21.51 | 30.36 |
| RAS | 6 | 12.5% | 1.57 | 33.81 | 21.62 | 30.33 |
| RAS | 5 | 75.0% | 1.59 | 44.02 | 23.14 | 29.53 |
| RAS | 5 | 50.0% | 1.75 | 48.65 | 24.51 | 29.29 |
| RFlow | 4 | 100% | 1.79 | 61.92 | 27.42 | 28.45 |
| RAS | 5 | 25.0% | 1.94 | 51.92 | 25.67 | 29.06 |
| RAS | 5 | 12.5% | 1.99 | 53.24 | 26.04 | 28.94 |
| RFlow | 3 | 100% | 2.38 | 121.61 | 36.92 | 25.32 |

Table 11: Full experiment results of *RAS* and rectified flow on Stable Diffusion 3 and COCO Val2014 1024×1024.

A.5    QUESTIONNAIRE FOR HUMAN EVALUATION

This section contains the questionnaire we used for the human evaluation we mentioned in Section 4.

**Text-to-Image Quality Preference Survey**

We are conducting an evaluation of two image generation methods. You will be presented with 14 pairs of images, each created by one of the two methods, with the order of the images shuffled for objectivity. Please select your preference for the shown images. Thank you for your participation and cooperation.

**Q1.** A massive alien spaceship that is shaped like a pretzel.

A.

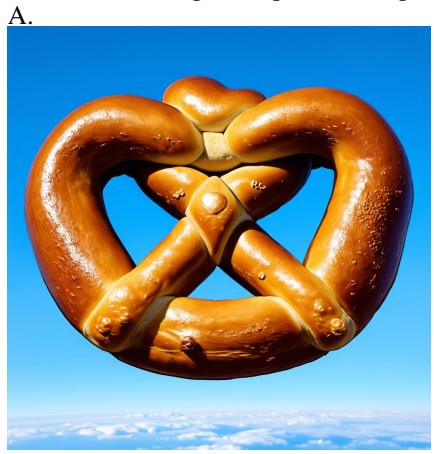

B.

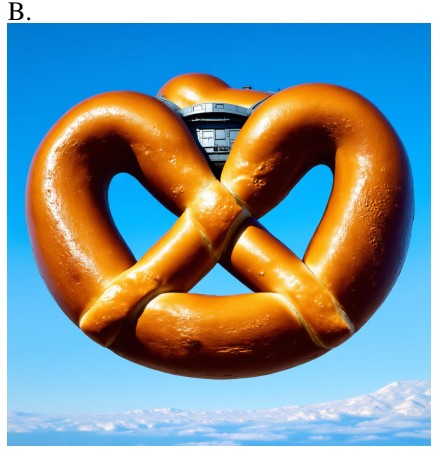

    ☐  A is obviously better than B.

    ☐  A is slightly better than B.

    ☐  They are of similar qualities.

    ☐  B is slightly better than A.

    ☐  B is obviously better than A.

**Q2.** Upper body of a young woman in a Victorian-era outfit with brass goggles and leather straps. Background shows an industrial revolution cityscape with smoky skies and tall, metal structures.

A.

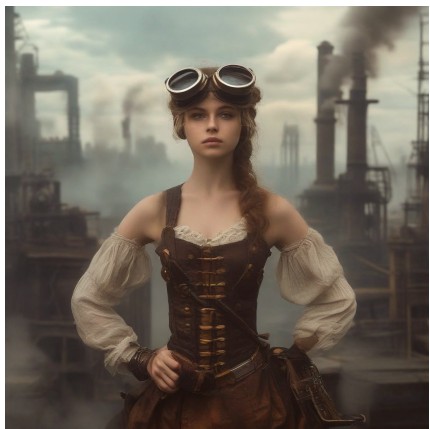

B.

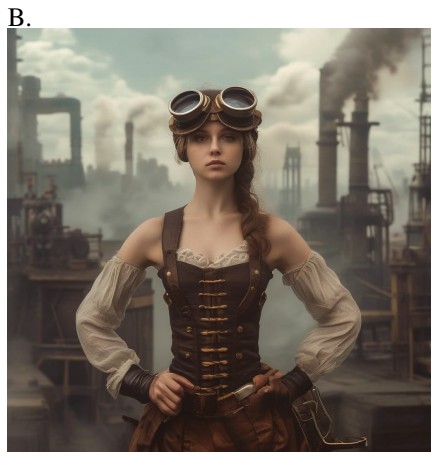

☐ A is obviously better than B.

☐ A is slightly better than B.

☐ They are of similar qualities.

☐ B is slightly better than A.

☐ B is obviously better than A.

**Q3.** This dreamlike digital art captures a vibrant, kaleidoscopic bird in a lush rainforest.

A.

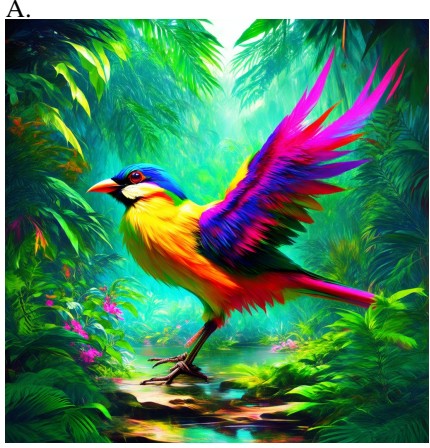

B.

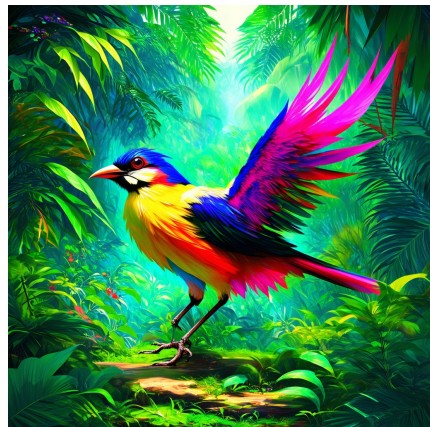

☐ A is obviously better than B.

☐ A is slightly better than B.

☐ They are of similar qualities.

☐ B is slightly better than A.

☐ B is obviously better than A.

**Q4.** A cat wearing a cowboy hat and sunglasses and standing in front of a rusty old white spaceship at sunrise. Pixar cute. Detailed anime illustration.

A.

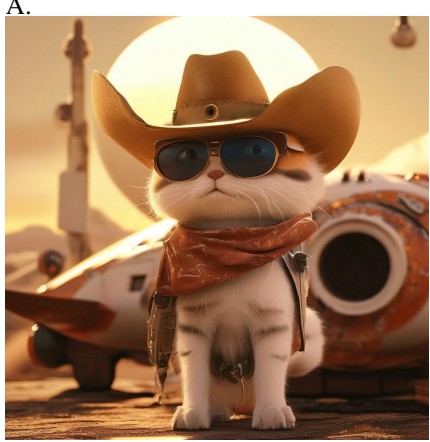

B.

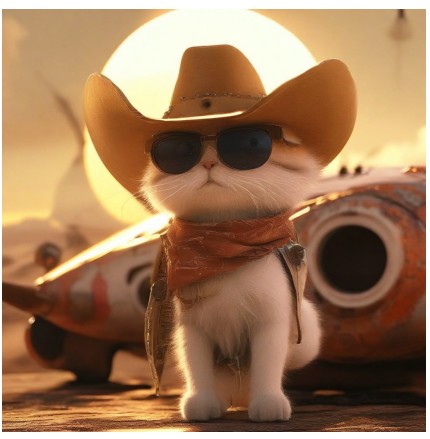

☐ A is obviously better than B.
☐ A is slightly better than B.
☐ They are of similar qualities.
☐ B is slightly better than A.
☐ B is obviously better than A.

**Q5.** A kangaroo holding a beer, wearing ski goggles and passionately singing silly songs.

A.

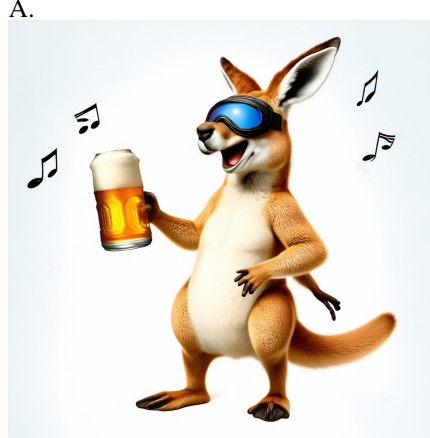

B.

☐ A is obviously better than B.

☐ A is slightly better than B.

☐ They are of similar qualities.

☐ B is slightly better than A.

☐ B is obviously better than A.

**Q6.** A photorealistic image of a Pagani Huayra driving through a city at night with glowing city lights in the background.

A.

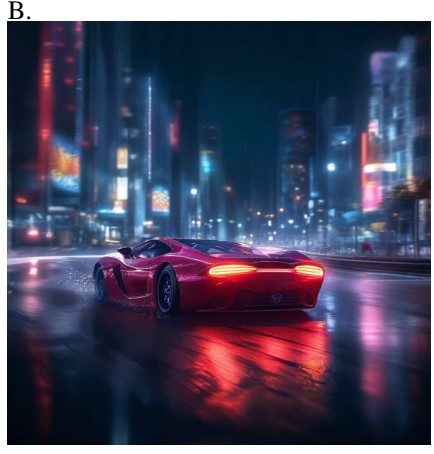

B.

☐ A is obviously better than B.
☐ A is slightly better than B.
☐ They are of similar qualities.
☐ B is slightly better than A.
☐ B is obviously better than A.

**Q7.** A cheeseburger with juicy beef patties and melted cheese sits on top of a toilet that looks like a throne and stands in the middle of the royal chamber.

A.

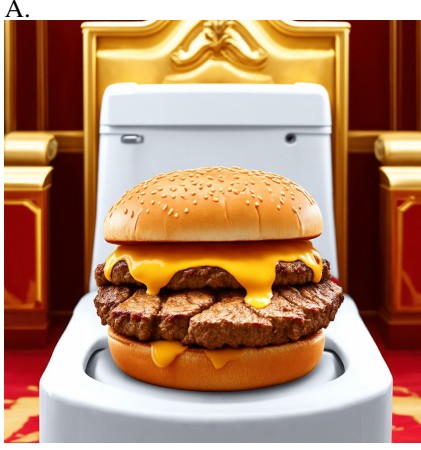

B.

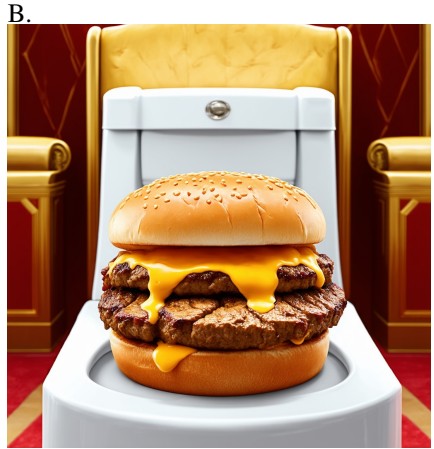

☐ A is obviously better than B.

☐ A is slightly better than B.

☐ They are of similar qualities.

☐ B is slightly better than A.

☐ B is obviously better than A.

**Q8.** A detailed photorealistic image of a steampunk locomotive on a platform with sharp lines, surrounded by light purple fog.

A.

B.

☐ A is obviously better than B.
☐ A is slightly better than B.
☐ They are of similar qualities.
☐ B is slightly better than A.
☐ B is obviously better than A.

**Q9.** An entire universe inside a bottle sitting on the shelf at Walmart on sale.

A.

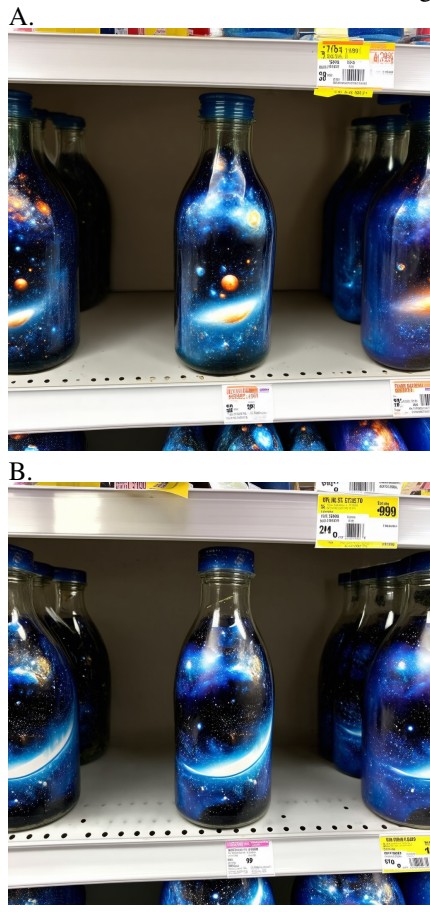

B.

☐ A is obviously better than B.

☐ A is slightly better than B.

☐ They are of similar qualities.

☐ B is slightly better than A.

☐ B is obviously better than A.

**Q10.** Snow-covered mountains reflected in a crystal-clear alpine lake, with a small wooden cabin nestled among tall pine trees.

A.

B.

☐ A is obviously better than B.
☐ A is slightly better than B.
☐ They are of similar qualities.
☐ B is slightly better than A.
☐ B is obviously better than A.

**Q11.** A red heart in the clouds over water, in the style of zbrush, light pink and sky-blue, I can't believe how beautiful this is, hyperbolic expression, nyc explosion coverage, unreal engine 5, robert bissell.

A.

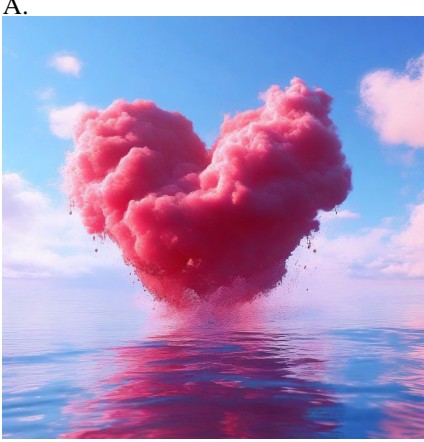

B.

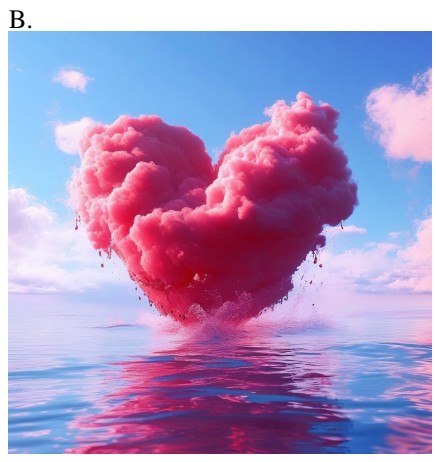

☐ A is obviously better than B.

☐ A is slightly better than B.

☐ They are of similar qualities.

☐ B is slightly better than A.

☐ B is obviously better than A.

**Q12.** Upper body of a young woman adorned in elaborate ancient Egyptian clothing, with a headdress featuring golden ornaments and colorful gemstones.The background shows the inside of a grand temple with hieroglyphics on the walls.

A.

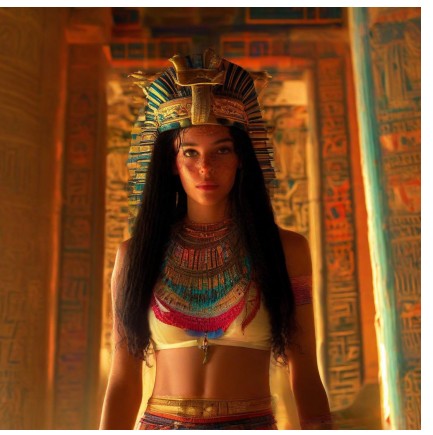

B.

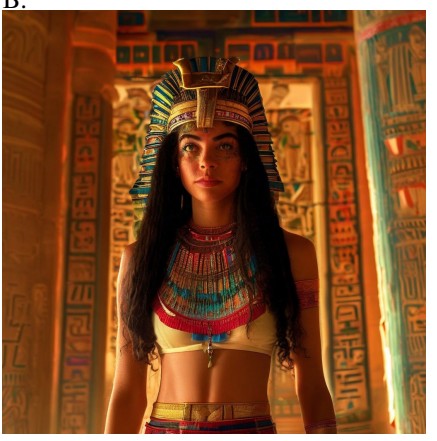

☐ A is obviously better than B.
☐ A is slightly better than B.
☐ They are of similar qualities.
☐ B is slightly better than A.
☐ B is obviously better than A.

**Q13.** The Hulk is in a colorful gothic background, with highly detailed dramatic lighting and a photo realistic style, rendered in 8K resolution.

A.

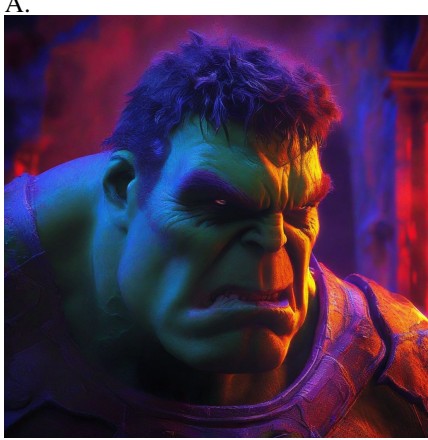

B.

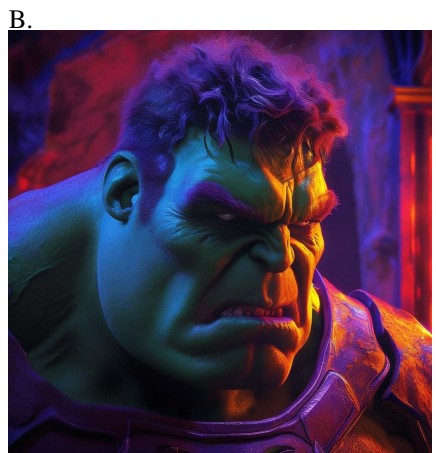

☐ A is obviously better than B.

☐ A is slightly better than B.

☐ They are of similar qualities.

☐ B is slightly better than A.

☐ B is obviously better than A.

**Q14.** A car made out of vegetables.

A.

B.

☐ A is obviously better than B.
☐ A is slightly better than B.
☐ They are of similar qualities.
☐ B is slightly better than A.
☐ B is obviously better than A.

