# OpenReview forum: "Region-Adaptive Sampling for Diffusion Transformers"
_ICLR.cc/2026/Conference — ICLR 2026 Conference Withdrawn Submission_

### Official Review · Reviewer_2sgM · 2025-10-25

**Soundness:** 3
**Presentation:** 3
**Contribution:** 3
**Rating:** 6
**Confidence:** 5

**Summary:**

This paper proposes a training-free sampling strategy that dynamically assigns different update ratios to image regions based on model attention, thereby improving generation efficiency while preserving output quality. Additionally, the authors carefully design specific strategies to eliminate the error accumulation in denoising process with the proposed method. Experiments conducted on Stable Diffusion 3 and Lumina-Next-T2I demonstrate the proposed method achieves up to 2.36× and 2.51× speedups, respectively, with minimal quality degradation.

**Strengths:**

1.	The insight presented in this paper is both novel and thought-provoking, and the experimental results demonstrate strong and consistent performance across various settings.

2.	The proposed method is simple yet effective. Its straightforward and elegant design ensures ease of implementation and makes it readily applicable to real-world scenarios.

3.	The approach is grounded in clear and well-motivated observations about redundancy and regional variation in diffusion models. This strong conceptual foundation renders the method not only empirically convincing but also theoretically sound. The analysis of token dynamics across timesteps adds valuable depth and supports the rationale behind the design choices.

4.	The paper is clearly written and well-organized, making the methodology and findings easy to follow. The overall presentation is professional and accessible, which facilitates understanding for a broad audience.

**Weaknesses:**

1. Heuristic Nature and Limited Theoretical Justification.
While the proposed method is empirically effective, its design is primarily heuristic. The use of noise standard deviation as an indicator of regional importance lacks strong theoretical grounding or formal justification. This raises concerns about the generalizability of the approach to different architectures, datasets, or diffusion formulations.

2. Sensitivity to Hyperparameters and Stability Issues.
The effectiveness of the proposed Region-Adaptive Sampling (RAS) relies on several critical hyperparameters, such as the sampling ratio, the smoothing factor for patch-level statistics, and the starvation penalty coefficient. However, the paper does not provide a systematic analysis of their influence. The potential sensitivity of these parameters may affect both stability and reproducibility in practical deployments.

3. Lack of Semantic Awareness in Importance Estimation.
The current method determines regional importance solely based on noise variance, which does not necessarily correlate with semantic significance. As a result, the model might focus excessively on low-level texture regions while overlooking semantically meaningful areas, particularly in multimodal or text-guided generation tasks.

4. Optimization Restricted to Intra-step Efficiency.
The proposed method reduces computation per diffusion step but does not fundamentally shorten the diffusion trajectory or reduce the total number of sampling steps. Consequently, the overall acceleration remains bounded by the inherent diffusion process length.

**Questions:**

None.

---

### Official Review · Reviewer_6Ct9 · 2025-10-27

**Soundness:** 2
**Presentation:** 1
**Contribution:** 2
**Rating:** 2
**Confidence:** 4

**Summary:**

This paper introduces a new sampling method called Region-adaptive sampling, which dynamically assigns different update ratios to image regions based on model foucs. RAS can accelerate the sampling speed up to 2.36x and 2.51x on SD3 and Lumina-Next-T2I, respectively, but almost no improvement on the generation quality.

**Strengths:**

This paper proposes a new sampling method on diffusion, called RAS. RAS aims to dynamically update the image reigons during the denoising steps based on the deviation as the metric. That menas that RAS can help the diffusion model concentrate on semantically meaningful areas, and reuse cached noise for others.

The idea seems interesting, however, it misses a lot of necessary details.  Please see the weaknesses.

**Weaknesses:**

I have carefully read this paper, and the writing issues are significant.

1. the mis-use of the format of reference. I think the authors mis-use the \citep and \citet. The reference format in this paper is totally wrong, making it difficult for me to recognize the main content.

2. writing typos. Line 107, "As shown in Figure 5." ; Line 130 and 146 "anda". Caption of Table 2, "Full experiment results are available in Figure 2".

3. missing reference. Line 215, "Layernorm and MLP".

4. non-standard variable definitions and disorganized formatting. There is common sense on the definition of the variables in diffusion models, like predicted noise $\epsilon$, but in this paper the author use $N$. Besides, the orders of the figures are disorganized. For exmaple, the Figure 4 and Figure 3 are referenced after Figure 8, making it hard to read.

Besides the writing issues, here are some technical ones.

1. no comparison methods. I think RAS is a method for accelerating sampling. The authors conduct the experiments on SD3 and Lumina-Next-T2I, and just compare with the standard sampling.

[1] PFDiff: Training-Free Acceleration of Diffusion Models Combining Past and Future Scores
[2] Training-free Diffusion Acceleration with Bottleneck Sampling

2. lack of the motivation. The motivation of this paper is weak, honestly. This phenomenon in Figure 3 has been mentioned in numerous papers. Besides, why the authors use the deviation of the model's output as the metric? The authors need to clarify it.

3. Through the experiments in this paper, I believe this is an acceleration method rather than an enhancement of generation. One awkward point is that RAS seems to achieve comparable quality to standard methods only at lower step counts, while outperforming them in speed. However, under such conditions, the quality of the standard method is largely unguaranteed—meaning the generated images will be of poor quality, such as at 5 steps. Conversely, RAS's performance at higher step counts falls significantly below that of standard methods, which greatly diminishes the practical value of t

**Questions:**

See the weaknesses. From my perspective, I to decline this paper. A major revision is needed.

---

### Official Review · Reviewer_ea17 · 2025-10-31

**Soundness:** 4
**Presentation:** 1
**Contribution:** 2
**Rating:** 2
**Confidence:** 4

**Summary:**

This paper introduces Region-Adaptive Sampling (RAS), a novel, training-free sampling strategy designed to accelerate inference in Diffusion Transformers (DiTs). The core idea is to exploit the observation that DiTs focus on semantically meaningful regions of an image, and these regions exhibit strong continuity across consecutive sampling steps. RAS dynamically allocates computational resources by assigning different update ratios to various image regions based on the model's focus.

**Strengths:**

1. The idea of exploiting spatial heterogeneity in the sampling process is straightforward and has the potential to be highly impactful.
2. A significant advantage of RAS is that it is a training-free method. This makes it easy to apply to a wide range of pre-trained diffusion models without the need for costly retraining.
3. The authors provide a set of experiments on state-of-the-art Diffusion Transformers. The reported speedups are substantial, and the quality of the generated images is well-maintained.

**Weaknesses:**

1. The manuscript's clarity is a major concern. The writing is difficult to follow, with grammatical errors and awkward phrasings. I strongly recommend a thorough revision of the entire manuscript by a native English speaker or a professional editing service.
2. More experiments about comparison with existing works are needed.

**Questions:**

see Weaknesses

---

### Official Review · Reviewer_3ba8 · 2025-11-01

**Soundness:** 3
**Presentation:** 3
**Contribution:** 3
**Rating:** 4
**Confidence:** 4

**Summary:**

The paper proposes Region-Adaptive Sampling (RAS), a training-free sampling strategy for Diffusion Transformers (DiTs) that dynamically updates only a subset of latent tokens at each diffusion step. RAS ranks tokens (patches) by a noise-based metric (e.g., std or ℓ2 norm of predicted noise), keeps frequently-dropped tokens’ previous noise cached, and only forwards the fast-update regions through the DiT while using cached keys/values for attention recovery; it also uses dynamic scheduling and periodic dense resets plus a starvation-prevention term to avoid drift. The method is implemented with kernel fusing for efficiency. Experiments on Stable Diffusion 3 and Lumina-Next-T2I (COCO) show up to ~2.4–2.5× throughput gains with small degradation in FID/CLIP and mixed-but-acceptable human preferences.

**Strengths:**

1. **Novel and practical idea exploiting DiT properties.** The paper leverages a property unique to transformer-based diffusion backbones (variable token handling) to enable spatially non-uniform sampling — a conceptually simple but impactful direction compared to uniform timestep reduction.

2. **Well-engineered, end-to-end system.** RAS contains multiple pragmatic components (metric, starvation prevention, error resets, KV caching, kernel fusing) that together address obvious failure modes, suggesting strong engineering maturity and deployability.

3. **Comprehensive experiments and ablations.** Experiments cover two modern DiT models (Stable Diffusion 3, Lumina-Next-T2I), a large COCO-based benchmark, multiple metrics (FID, sFID, CLIP), ablations that demonstrate necessity of components (drop scheduling, KV caching, reset schedule, starvation prevention), and human studies reporting preference distributions and benchmark scores. This breadth supports the claims.

4. **Clear quantifiable speed–quality gains.** The paper presents Pareto improvements vs. rectified flow: substantial throughput gains (e.g., ~1.6× in human evaluation, up to ~2.4–2.5× in other configs) often with only modest quality loss, and sometimes with better FID/CLIP at similar throughputs. These are meaningful for real-time or constrained settings.

**Weaknesses:**

1. **Comparisons to strongest baselines could be deeper.** The baseline is rectified flow (uniform timestep reduction) and some cached-layer methods; however, recent fast-sampling / distillation or scheduler adaptations (and combined methods) may offer competitive alternatives. It is unclear if RAS can be combined with all such methods or whether combined evaluation was performed. The paper claims orthogonality but empirical combination results are limited.

2. **Kernel detail.** The paper proposes an optimization on the kernel, but there are no quantitative results demonstrating the actual effect of the kernel. Providing quantitative improvements after the optimization would further enhance the quality of the paper.

3. **Clarity and reproducibility details missing in places.** While many implementation details are given, some choices (precise hyperparameters for metric thresholds, scheduling of dense resets across different step counts, how the exp(k·D) coefficient k was chosen) are not exhaustively described in the main text and would be needed for exact reproduction. The appendix helps but could be clearer.

**Questions:**

1. **Sensitivity to metric & hyperparameters:** How sensitive is RAS to the choice of metric (std vs. ℓ2 norm) and to the scale factor k in the starvation term? Can you provide guidance or automated tuning rules for k and the sampling-ratio schedule for new models/datasets?

2. **Failure cases & visual examples:** For the example presented in Figure 7, there is a significant loss of detail in the results generated by RAS. Is there a natural methodological disadvantage to RAS such spatial perception?

3. **Memory & device constraints:** You report a 4–6% extra memory overhead on large A100s; how does memory overhead scale with resolution, patch size, and model size? What are the implications for common GPU setups with <24 GB?

---

### Note · Authors · 2025-11-12

I have read and agree with the venue's withdrawal policy on behalf of myself and my co-authors.